# Optic Flow Speed and Retinal Stimulation Influence Microsaccades

**DOI:** 10.3390/ijerph19116765

**Published:** 2022-06-01

**Authors:** Milena Raffi, Aurelio Trofè, Andrea Meoni, Luca Gallelli, Alessandro Piras

**Affiliations:** 1Department of Biomedical and Neuromotor Sciences, University of Bologna, 40126 Bologna, Italy; andrea.meoni@unibo.it (A.M.); alessandro.piras3@unibo.it (A.P.); 2Department of Quality of Life, University of Bologna, 47921 Rimini, Italy; aurelio.trofe2@unibo.it; 3Department of Health Science, School of Medicine, University of Catanzaro, 88100 Catanzaro, Italy; gallelli@unicz.it; 4Clinical Pharmacology and Pharmacovigilance Unit, Mater Domini Hospital Catanzaro, 88100 Catanzaro, Italy

**Keywords:** self-motion perception, visual perception, visual processing, eye position, eye movements, sensorimotor control, attention, visual system

## Abstract

Microsaccades are linked with extraretinal mechanisms that significantly alter spatial perception before the onset of eye movements. We sought to investigate whether microsaccadic activity is modulated by the speed of radial optic flow stimuli. Experiments were performed in the dark on 19 subjects who stood in front of a screen covering 135 × 107° of the visual field. Subjects were instructed to fixate on a central fixation point while optic flow stimuli were presented in full field, in the foveal, and in the peripheral visual field at different dot speeds (8, 11, 14, 17, and 20°/s). Fixation in the dark was used as a control stimulus. For almost all tested speeds, the stimulation of the peripheral retina evoked the highest microsaccade rate. We also found combined effects of optic flow speed and the stimulated retinal region (foveal, peripheral, and full field) for microsaccade latency. These results show that optic flow speed modulates microsaccadic activity when presented in specific retinal portions, suggesting that eye movement generation is strictly dependent on the stimulated retinal regions.

## 1. Introduction

Eye movements are used to scan the environment to select visual stimuli, so they are critically important in determining what we see and attend to [1,2,3,4,5,6,7,8,9,10]. Microsaccades are small eye movements produced during attempted fixations. In past decades, studies on microsaccades have successfully demonstrated their role in contrasting visual fading [11,12], driving attentional and discrimination tasks [13,14,15,16,17], and processing the visual field [18,19,20,21,22].

The optic flow field is a crucial input to navigate the environment. Several neurophysiological studies demonstrated specific neuronal sensitivity to optic flow in some cortical and subcortical areas of the monkey’s brain [23,24,25,26,27,28,29,30,31,32,33,34,35]. Some studies showed optic flow sensitivity in various areas of the human brain, such as areas MST and V3A [36], area MT [37] and the fronto-temporo-parietal network [38].

Speed represents an important input while navigating the environment. Neuronal sensitivity to speed has been demonstrated in areas 7a and MST in the monkey [39,40]. Several studies on human subjects report that many motor actions such as steering or walking are influenced by the speed of optic flow patterns. Kountouriotis et al. [41] showed that the human brain performs global averaging of optic flow speed from the visual field and uses this signal as an input for steering control. Durgin and Gigone [42] demonstrated that the discrimination thresholds for appropriate visual speeds are enhanced during walking. Gait parameters are modulated by optic flow speed; when optic flow speed decreases, walking speed and stride length increase [43,44]. The motor response to heading perception is also modulated by the speed of optic flow patterns [45].

In a previous study, we used optic flow stimuli to reveal the relationship between heading perception and microsaccades in a discrimination task. The results demonstrated that microsaccades’ characteristics and directions are related to heading perception [20]. In the present work, we moved on to analyze the microsaccades’ characteristics during the view of radial optic flow stimuli with different dot speeds. The rationale for this study arises from evidence that during daily life, our gaze explores the visual field while we move in the environment at different speeds of motion; i.e., we can walk slowly in a downtown street while looking at stores, we can walk quickly in a busy road while scanning the environment to cross the street, and we can run in the park while looking straight ahead to keep the right heading. During walking, people discover the relationship among the magnitudes of optic flow produced by varying magnitudes of walking [46]. All of our daily behaviors change the optic flow fields depending on the task we are performing [47,48,49]; thus, our nervous system continuously shifts from covert to overt attention [50,51,52]. Given that microsaccades seem to indicate where we unconsciously focus our attention [53,54], we aimed to verify whether the speed of the optic flow stimulus could directly influence microsaccades’ generation and properties.

## 2. Materials and Methods

Experiments were performed on 19 healthy volunteers, 9 females and 10 males, who had normal or corrected to normal vision. The subjects’ ages ranged from 21 to 35 years (average 27.9 ± 4.2 SD). Before the beginning of the experiment, each participant signed a written informed consent form. The study protocol was approved by the Institutional Bioethic Committee of the University of Bologna. The experiments were performed in accordance with the ethical standards laid down in the 1964 Declaration of Helsinki.

To assess hand and foot laterality, each participant completed a laterality test, which is a revised version of the standardized Waterloo Footedness Questionnaire (WFQ) and Waterloo Handedness Questionnaire (WHQ) [55,56]:[(right preference − left preference)/(right preference + left preference)] × 100

A positive index indicates right dominance, while a negative index indicates left dominance. The cut-off points to determine the degree of handiness of the participants were 70 for right-handed and −70 for left-handed. The rationale for computing a laterality index was to correlate the microsaccades’ directions with the dominant side in order to elucidate the potential mechanisms for motor control.

### 2.1. Optic Flow Stimuli

In this experiment, we presented three types of optic flow stimuli that stimulated different portions of the visual field: foveal, full, and peripheral (Figure 1A–C). In this study, we used the same stimuli described in a previous paper [57]. In brief, we considered the foveal visual field as the 7° surrounding the fovea. We considered the periphery as the visual field outside the inner 20° of the foveal visual field. The stimuli were generated by white dots with a luminous intensity of 1.3 cd/m^2^ and a size of 0.4°. The stimuli were back-projected onto a translucent screen that covered 135 × 107° of the visual field. All recordings were performed in a dark room with dark walls. Each participant was instructed to look at a fixation point (FP) of 0.6° in size without moving the head. The screen height was adjusted for each subject to have the FP in the primary position. The subjects were positioned 115 cm away from the screen. To assess the dependence of microsaccades on the velocity of the stimulus, we varied the dot speed in all three stimuli, thereby obtaining 15 different conditions; the tested speeds of the optic flow stimulus were: 8°/s, 11°/s, 14°/s, 17°/s, and 20°/s. A control stimulus (baseline) consisted of simple fixation on a dark screen (Figure 1D). Optic flow stimuli were produced using Matlab psychophysical toolbox (The Mathworks Inc., Natick, MA, USA). We recorded 2 repetitions for both the baseline and the optic flow stimuli with different speeds; thus, each subject performed 32 trials. Each trial lasted 30 s.

### 2.2. Eye Movements and Eye Position Recordings

In this experiment, horizontal and vertical eye movements were recorded using an EyeLink video-based eye tracking system (EyeLink^®^ II, SR Research Ltd., Mississauga, ON, Canada), which consists of two miniature cameras mounted on a leather-padded headband. Pupil tracking was recorded at 500 Hz, with high spatial resolution (<0.005°) and low noise (<0.01°). At the beginning of each recording session, we calibrated the eye tracking; we instructed the subject to fixate on a target presented in a random order in a nine-point 25 × 25° square grid. After correct camera calibration, the data were validated, and drift correction was executed by applying a corrective offset to the raw eye-position.

### 2.3. Data Analysis

Microsaccades are defined as small eye movements that occur during prolonged visual fixation with an amplitude of less than 1° [58]. To identify microsaccades, we developed an algorithm based on that of Otero-Millan [59], and it has already been used in a previous study [60]. To reduce the amount of potential noise, we considered only binocular microsaccades for at least 3 data samples (6 ms). Trials with incorrect fixations, eye blinks, or behavioral errors were discarded. Portions of data with very fast decreases and increases in the pupil area (>50 units/sample) were removed from the analysis, given that such periods are likely semi-blinks where the pupil is never fully occluded. We also ignored the 200 ms before and after each blink/semi-blink to eliminate the initial and final periods when the pupil was still partially occluded [61].

We computed the amplitude, duration, latency, rate and peak velocity of microsaccades for each subject in each trial and in each condition. Then, we averaged the values for all subjects in each condition and trial. We computed the microsaccade rates considering only time spent in fixation periods; the total number of microsaccades for each subject in each trial was divided by the total time spent in fixation during that specific trial. We computed the microsaccades’ latency by averaging all latency from the trial’s onset.

Statistical analysis was performed using a repeated-measures ANOVA with stimuli (foveal, full field, and periphery) and speed (8, 11, 14, 17, and 20°/s) as within-subject factors and with the baseline as covariant (SPSS^®^ 22.0, Chicago, IL, USA). Results were considered significant at *p* < 0.05.

## 3. Results

The analysis of the laterality test showed that 16 participants were strongly right-handed as values were above 78. Three subjects were left-handed, and their values were −44, −55, and −89, indicating a strong left laterality for only one subject.

The analysis of the repeated measures ANOVA showed that the microsaccades’ peak velocity, latency, and rate showed significant effects, while the microsaccades’ amplitude and duration did not show any significant effect. 

### 3.1. Microsaccades Peak Velocity

The ANOVA performed on the microsaccades’ peak velocity showed a main effect of stimulus (F_1,2_ = 3.676, *p* = 0.037, η_p_^2^ = 0.197) and an interaction effect of stimulus × baseline (F_1,2_ = 4.601, *p* = 0.018, η_p_^2^ = 0.235). Figure 2 shows the changes in the microsaccades’ peak velocity across stimuli, and it shows that the baseline differed from the three optic flow stimuli.

The non-significant effects values were: speed (F_1,4_ = 1.857, *p* = 0.130, η_p_^2^ = 0.110), speed × baseline (F_1,4_ = 1.771, *p* = 0.146, η_p_^2^ = 0.106), speed x stimulus (F_1,8_ = 0.469, *p* = 0.876, η_p_^2^ = 0.030), and speed × stimulus × baseline (F_1,8_ = 0.480, *p* = 0.869, η_p_^2^ = 0.031).

### 3.2. Microsaccades Latency

The ANOVA performed on the microsaccades’ latency showed an interaction effect of stimulus × speed (F_1,8_ = 2.842, *p* = 0.006, η_p_^2^ = 0.159), indicating a combined effect of optic flow speed and the stimulated retinal region. The results also show an interaction effect of stimulus × speed × baseline (F_1,8_ = 2.734, *p* = 0.008, η_p_^2^ = 0.154), indicating that the combined effect of optic flow speed and the stimulated retinal region differed from the baseline. The Bonferroni pairwise comparison showed the following differences: baseline vs. 11°/s P, *p* = 0.027; baseline vs. 20°/s P, *p* = 0.001; 8°/s FO vs. 8°/s FU; *p* = 0.040; 8°/s FO vs. 20°/s P, *p* = 0.002; 8°/s FU vs. 11°/s P, *p* = 0.002; 8°/s P vs. 11°/s P, *p* = 0.024; 8°/s P vs. 20°/s P, *p* = 0.023; 11°/s FU vs. 11°/s P, *p* = 0.009; 11°/s FO vs. 11°/s P, *p* = 0.009; 11°/s P vs. 14°/s FO, *p* = 0.006; 11°/s P vs. 17°/s FO, *p* = 0.009; 11°/s P vs. 17°/s FU, *p* = 0.044; 11°/s P vs. 17°/s P, *p* = 0.012; 11°/s P vs. 20°/s FO, *p* = 0.031; 11°/s P vs. 20°/s FU, *p* = 0.004; 11°/s P vs. 20°/s P, *p* < 0.001; 14°/s FU vs. 20°/s P, *p* = 0.007; 17°/s FU vs. 20°/s P, *p* = 0.048. Figure 3 shows microsaccade latency in all conditions. The microsaccades’ latency differed within and across stimuli.

The non-significant effects values were: speed (F_1,4_ = 0.400, *p* = 0.808, η_p_^2^ = 0.026), speed × baseline (F_1,4_ = 0.487, *p* = 0.745, η_p_^2^ = 0.031), stimulus (F_1,2_ = 0.525, *p* = 0.597, η_p_^2^ = 0.034), and stimulus × baseline (F_1,2_ = 0.525, *p* = 0.597, η_p_^2^ = 0.034).

### 3.3. Microsaccades Rate

The ANOVA performed on microsaccade rate showed a main effect of stimulus (F_1,2_ = 4.774, *p* = 0.016, η_p_^2^ = 0.241) and an interaction effect of speed × stimulus × baseline (F_1,8_ = 2.623, *p* = 0.011, η_p_^2^ = 0.149), indicating that the combined effect of optic flow speed and the stimulated retinal region differed from the baseline. The Bonferroni pairwise comparison showed the following differences: baseline vs. 8°/s FO°/s, *p* = 0.043; baseline vs. 8°/s FU, *p* = 0.041; baseline vs. 11°/s FO, *p* = 0.014; 8°/s P vs. 11°/s FO, *p* = 0.029; 11°/s FO vs. 14°/s P, *p* = 0.039. Figure 4 shows that the stimulation of the peripheral retina evoked a higher microsaccade rate.

The non-significant effects values were: speed (F_1,4_ = 1.718, *p* = 0.158, η_p_^2^ = 0.103), speed × baseline (F_1,4_ = 1.498, *p* = 0.214, η_p_^2^ = 0.091), stimulus × baseline (F_1,2_ = 3.076, *p* = 0.061, η_p_^2^ = 0.170), and speed × stimulus (F_1,8_ = 1.526, *p* = 0.155, η_p_^2^ = 0.092).

## 4. Discussion

When we move in the environment, we precisely evaluate our heading and speed. Speed information can be obtained by integrating visual speed from optic flow and locomotion. Given the importance of optic flow for navigation in the environment and the involvement of microsaccades in visual processing, we sought to investigate whether microsaccadic activity is modulated by the speed of radial optic flow stimuli. In this study we decided to test five speeds (8, 11, 14, 17, and 20°/s), starting from 8°/s, which represents a slow walk, and ending with 20°/s, which represents a fast run. The results show that the microsaccades’ latency and rate were influenced by the speed of radial optic flow stimuli in combination with the stimulated retinal region.

Microsaccade rate is linked to cognitive load in several neural processes. Microsaccade rate is modulated by the load of the working memory [62] and by action preparation [63,64,65]. During mental arithmetic processing, task difficulty has been shown to reduce microsaccade rate [66,67]. Valsecchi and colleagues [68,69] showed that a reduced microsaccade rate might be related to memory updating. Betta and Turatto [70] found a strong reduction in microsaccade rate during the preparation of a manual response. Piras et al. [15] performed an experiment on goalkeepers who needed to predict the ball direction of a penalty kick. The results of that study showed that the temporal sequence of microsaccade rates decreased by ~1000 ms just before the goalkeepers’ final movement initiation. There is a general consensus about the inverse relationship between microsaccade rate and cognitive load. Our results indicate that microsaccade rate is influenced by both the optic flow speed and the stimulated retinal region. Microsaccade rate is significantly higher in the peripheral stimulation at almost all tested speeds. This result could be interpreted in the framework of the stabilizing effect that stimulation of the peripheral retina has on postural control. While it must be noted that the literature is somewhat controversial, several studies agree with the view that stimulation of the peripheral visual field stabilizes posture (cfr. [71] for review), so it is possible to hypothesize that such a stabilizing effect requires a lighter cognitive load allowing the generation of more microsaccades in a search strategy.

The present results are also in line with those of previous perceptual studies. It has been shown that when a cue is shown in a peripheral portion of the visual field, the frequency of microsaccades in the cue’s direction increases [54,72]. On the other hand, when a cue is presented centrally, the frequency of microsaccades takes more time to move in the cue’s direction [53,73]. Although there was no specific attentional cue in the present study, the moving dots in various portions of the visual field were a powerful visual stimulus. Neuronal density and spatial resolution are low in the peripheral retina because it is specialized for movement detection and sensitivity; the cortical regions that process this information are also limited in size with respect to the processing of foveal information [74]. Peripheral visual function has a more pivotal role than that of central vision in locomotion and postural stability (cfr. [74] for review), so it is reasonable that the frequency of microsaccades increases with peripheral vision.

The latency of the microsaccades showed combined effects between optic flow speed and the stimulated retinal region. As evident from Figure 3, in some cases, microsaccades were more concentrated toward the end of the trial (i.e., 11°/s periphery, 14°/s periphery, and 8°/s fovea), while in some other cases, they occurred more at the beginning of the trial (i.e., 20°/s periphery and 8°/s full field). This lack of clear relationship between optic flow speed and stimulus could be due to the nature of the task. First, each trial lasted 30 s allowing the generation of several microsaccades (about 30–50 per trial). Second, the participants were instructed to keep their gaze on a central target, and the dots did not have any behavioral meaning. Although it is important to show and discuss this interaction effect between optic flow speed and the stimulated retinal region, we believe that more studies are needed to access the main effect of speed on latency. Future studies should be designed using very short trials that allow the recording of only one microsaccade per trial.

It is worth noting the lack of a significant main effect of optic flow speed; indeed, we found combined effects only. In this experiment, we instructed our subjects to keep their gaze on the fixation point while ignoring the optic flow field. As already stated by Hafed et al. [75], it is necessary to study microsaccades in experiments that require fixation to be able to make inferences about vision and cognitive processes. As shown by Wang et al. [76], the view of radial expanding optic flow stimuli attracts attention toward the focus of expansion. It is possible that the experimental conditions used in this study were not optimal to uncover stimulus speed modulation on microsaccades. A task designed with an attentional cue upon different optic flow speed may be helpful in determining the exact relationship between microsaccades and stimulus speed.

## 5. Conclusions

In this study, we showed that microsaccades are modulated by the speed of optic flow stimuli when presented in specific retinal regions. The peripheral retina plays a critical role in visual motion perception. It is thus possible to hypothesize that such high generation of microsaccades is used in heading perception strategies. All things considered, the present results seem to suggest a different involvement of retinal regions in triggering microsaccades.

## Figures and Tables

**Figure 1 ijerph-19-06765-f001:**
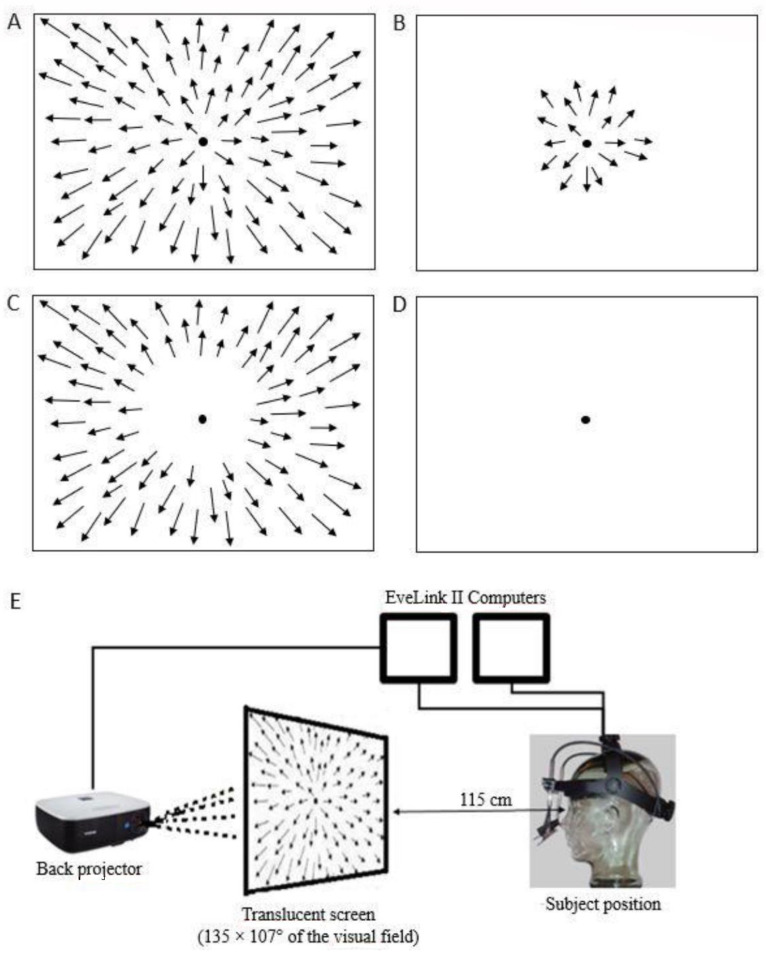
Optic flow and control stimuli. (**A**) Full field stimulation. (**B**) Foveal stimulation. (**C**) Peripheral stimulation. (**D**) Baseline condition (control). (**E**) Illustration of the experimental setup. Full, foveal, and peripheral stimuli were presented at different speeds: 8°/s, 11°/s, 14°/s, 17°/s, 20°/s. Arrows represent the velocity vectors of moving dots.

**Figure 2 ijerph-19-06765-f002:**
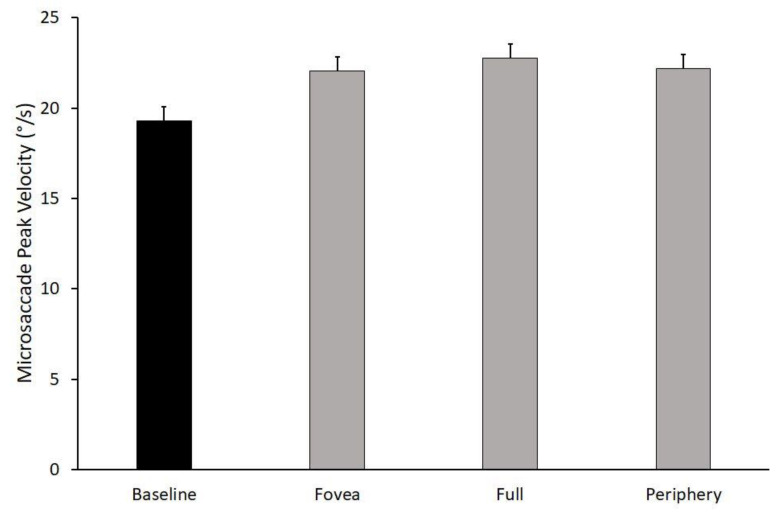
Frequency histograms of microsaccades peak velocity in all stimuli. Grey histograms represent optic flow stimuli, black histogram represents baseline. Data are reported as mean ± SE.

**Figure 3 ijerph-19-06765-f003:**
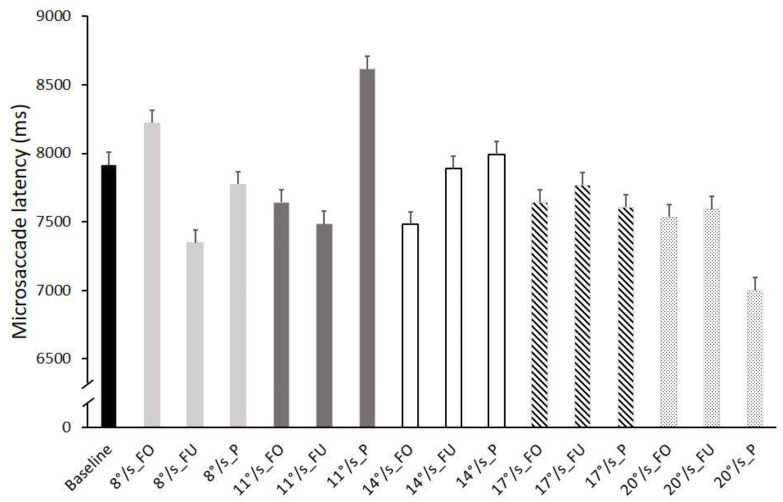
Frequency histograms of averaged microsaccade latency in all conditions. Black: baseline; light grey: speed 8°/s; dark grey: speed 11°/s; white: speed 14°/s; dashed lines: speed 17°/s; dotted lines: speed 20°/s; FO: fovea; FU: full field; P: periphery. Data are reported as mean ± SE.

**Figure 4 ijerph-19-06765-f004:**
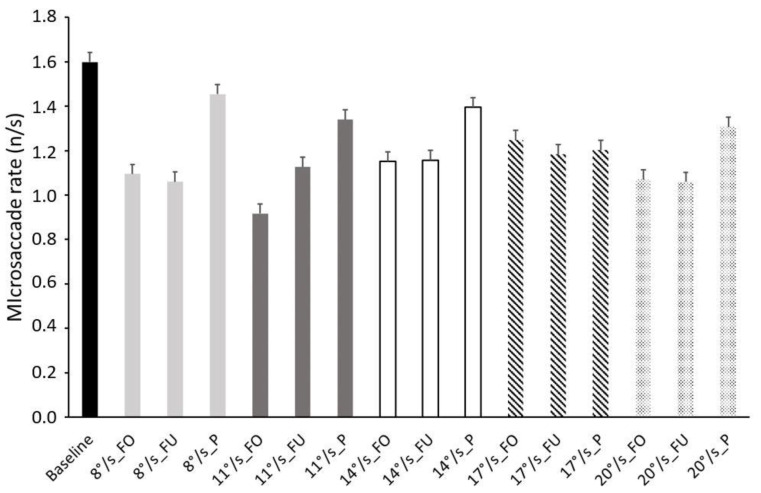
Frequency histograms of microsaccade rate in all conditions. Black: baseline; light grey: speed 8°/s; dark grey: speed 11°/s; white: speed 14°/s; dashed lines: speed 17°/s; dotted lines: speed 20°/s; FO: fovea; FU: full field; P: periphery. Data are reported as mean ± SE.

## Data Availability

Data sharing is not applicable to this article because of the consent provided by participants on the use of confidential data.

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
