# Peer review of "Optic Flow Speed and Retinal Stimulation Influence Microsaccades"

_ijerph, 2022, doi:10.3390/ijerph19116765_

Round 1

Reviewer 1 Report

The article describes issues related to microsaccades and proposes Optic flow speed and retinal stimulation influence microsaccades. The proposed research methodology allows to assess the effect of optic flow speed and stimulated retinal region for microsaccade latency. The results of these studies allow to determine the factors and their participation in the activity of microsaccades. In order to verify the assumptions made, the authors recruited 19 volunteers who participated in the research. After the theoretical part, the results of experimental research were presented in the paper and the comparative analysis was correctly compiled. The work is well organized with an introduction, theoretical part, application part and conclusions. Each part is represented correctly. The research results are significant and are correlated with the rest of the article.

Recommendations for the improvement of the manuscript:

  1. At what distance from the screen were the subjects placed?
  2. What was the size of the stimulation monitor?
  3. It is worth adding a diagram showing the research stand.
  4. Was the head of the examined person stabilized during the examination (placed on the chin)?
  5. Was the eye fixation additionally controlled in addition to the recommendations?
  6. Has the effect of incorrect eye fixation been investigated?
  7. There is no explanation to the notations F1,x, p, ηp2?
  8. What was the dictated choice of the stimulus speed?
  9. Could there be tiredness of the examined person?
  10. Could the tiredness have affected the test results?

Author Response

Reviewer 1

The article describes issues related to microsaccades and proposes Optic flow speed and retinal stimulation influence microsaccades. The proposed research methodology allows to assess the effect of optic flow speed and stimulated retinal region for microsaccade latency. The results of these studies allow to determine the factors and their participation in the activity of microsaccades. In order to verify the assumptions made, the authors recruited 19 volunteers who participated in the research. After the theoretical part, the results of experimental research were presented in the paper and the comparative analysis was correctly compiled. The work is well organized with an introduction, theoretical part, application part and conclusions. Each part is represented correctly. The research results are significant and are correlated with the rest of the article.

Recommendations for the improvement of the manuscript:

1. At what distance from the screen were the subjects placed?

The subjects were 115 cm away from the screen. This has been added at lines 89-90.

2. What was the size of the stimulation monitor?

It is written at line 86.

3. It is worth adding a diagram showing the research stand.

Figure 1 has been modified according to reviewer’s suggestion. See figure and legend at lines 97-99.

4. Was the head of the examined person stabilized during the examination (placed on the chin)?

The head was not stabilized. Subjects were instructed not to move the head. This has been added at line 88.

5. Was the eye fixation additionally controlled in addition to the recommendations?

The eye fixation was continuously monitored during the whole recording by a computer (illustrated in figure 1).

6. Has the effect of incorrect eye fixation been investigated?

We have not analyzed saccades made to peripheral locations, we only analyzed correct fixations.

7. There is no explanation to the notations F1,x, p, ηp2?

Those notations refer to ANOVA parameter, indicating the degree of freedom, the significant value and the error.

8. What was the dictated choice of the stimulus speed?

The idea was to test different speed starting from regular speed (8°/s) to fast speed (20°/s) at regular intervals of 3°/s.

9. Could there be tiredness of the examined person?

We do not think the subjects got tired as there was an interval between the two sessions. We allowed the subject to sit down as much as they liked between recordings.

10. Could the tiredness have affected the test results?

We do not think so. See answer to comment 9.

Reviewer 2 Report

This is a nice paper in which an experiment on microsaccades and optical flow is illustrated. The results show an intriguing link between optical flow, retinal characteristics, and the production of microsaccades.

I think this is a well-written and organized paper and is surely suitable for the special issue. I only have some minor comments and I hope these are useful to the authors.

Lines 69-75: Please clarify if you used a standardized questionnaire (e.g., Edinburgh Handedness Inventory) and the cut-off points to determine the degree of handiness of the participants.

Line 134: For completes I would also report a summary of the non-significant analyses (e.g., footnote?).

Lines 185-187: Other papers have shown a link between working memory/cognitive load/action preparation and microsaccades, and for completeness should be cited in this section (Dalmaso et al., 2017, Hermens et al., 2010; Watanabe et al., 2013; Xue et al., 2017).

References

Dalmaso, M., Castelli, L., Scatturin, P., & Galfano, G. (2017). Working memory load modulates microsaccadic rate. Journal of Vision, 17(3), 6. https://doi.org/10.1167/17.3.6

Hermens, F., Zanker, J. M., & Walker, R. (2010). Microsaccades and preparatory set: A comparison between delayed and immediate, exogenous and endogenous pro- and anti-saccades. Experimental Brain Research, 201(3), 489–498. https://doi.org/10.1007/s00221-009-2061-5

Watanabe, M., Matsuo, Y., Zha, L., Munoz, D. P., & Kobayashi, Y. (2013). Fixational saccades reflect volitional action preparation. Journal of Neurophysiology, 110(2), 522–535. https://doi.org/10.1152/jn.01096.2012

Xue, L., Huang, D., Wang, T., Hu, Q., Chai, X., Li, L., & Chen, Y. (2017). Dynamic modulation of the perceptual load on microsaccades during a selective spatial attention task. Scientific Reports, 7(1), 16496. https://doi.org/10.1038/s41598-017-16629-2

Author Response

Reviewer 2

This is a nice paper in which an experiment on microsaccades and optical flow is illustrated. The results show an intriguing link between optical flow, retinal characteristics, and the production of microsaccades.

I think this is a well-written and organized paper and is surely suitable for the special issue. I only have some minor comments and I hope these are useful to the authors.

We thank you very much for your comments and for your help in improving the quality of the paper.

Lines 69-75: Please clarify if you used a standardized questionnaire (e.g., Edinburgh Handedness Inventory) and the cut-off points to determine the degree of handiness of the participants.

We specified this at lines 70-76.

Line 134: For completes I would also report a summary of the non-significant analyses (e.g., footnote?).

Non-significant effects have been added at lines 144-146, 165-167, 182-184.

Lines 185-187: Other papers have shown a link between working memory/cognitive load/action preparation and microsaccades, and for completeness should be cited in this section (Dalmaso et al., 2017, Hermens et al., 2010; Watanabe et al., 2013; Xue et al., 2017).

The references have been added at lines 200-202.

References

Dalmaso, M., Castelli, L., Scatturin, P., & Galfano, G. (2017). Working memory load modulates microsaccadic rate. Journal of Vision17(3), 6. https://doi.org/10.1167/17.3.6

Hermens, F., Zanker, J. M., & Walker, R. (2010). Microsaccades and preparatory set: A comparison between delayed and immediate, exogenous and endogenous pro- and anti-saccades. Experimental Brain Research201(3), 489–498. https://doi.org/10.1007/s00221-009-2061-5

Watanabe, M., Matsuo, Y., Zha, L., Munoz, D. P., & Kobayashi, Y. (2013). Fixational saccades reflect volitional action preparation. Journal of Neurophysiology110(2), 522–535. https://doi.org/10.1152/jn.01096.2012

Xue, L., Huang, D., Wang, T., Hu, Q., Chai, X., Li, L., & Chen, Y. (2017). Dynamic modulation of the perceptual load on microsaccades during a selective spatial attention task. Scientific Reports7(1), 16496. https://doi.org/10.1038/s41598-017-16629-2
